# CNN-Based Facial Expression Recognition with Simultaneous Consideration of Inter-Class and Intra-Class Variations

**DOI:** 10.3390/s23249658

**Published:** 2023-12-06

**Authors:** Trong-Dong Pham, Minh-Thien Duong, Quoc-Thien Ho, Seongsoo Lee, Min-Cheol Hong

**Affiliations:** 1Department of Information and Telecommunication Engineering, Soongsil University, Seoul 06978, Republic of Korea; dongpt@ssu.ac.kr (T.-D.P.); duongthien2206@soongsil.ac.kr (M.-T.D.); hoquocthiendl@soongsil.ac.kr (Q.-T.H.); 2Department of Intelligent Semiconductor, Soongsil University, Seoul 06978, Republic of Korea; sslee@ssu.ac.kr; 3School of Electronic Engineering, Soongsil University, Seoul 06978, Republic of Korea

**Keywords:** facial expression recognition, convolutional neural networks, loss function, intra-class variations, inter-class variations

## Abstract

Facial expression recognition is crucial for understanding human emotions and nonverbal communication. With the growing prevalence of facial recognition technology and its various applications, accurate and efficient facial expression recognition has become a significant research area. However, most previous methods have focused on designing unique deep-learning architectures while overlooking the loss function. This study presents a new loss function that allows simultaneous consideration of inter- and intra-class variations to be applied to CNN architecture for facial expression recognition. More concretely, this loss function reduces the intra-class variations by minimizing the distances between the deep features and their corresponding class centers. It also increases the inter-class variations by maximizing the distances between deep features and their non-corresponding class centers, and the distances between different class centers. Numerical results from several benchmark facial expression databases, such as Cohn-Kanade Plus, Oulu-Casia, MMI, and FER2013, are provided to prove the capability of the proposed loss function compared with existing ones.

## 1. Introduction

Facial expressions have been used as critical and natural signals to represent human emotions and intentions. Therefore, various facial expression recognition (FER) methods have been studied and applied to fields, such as virtual reality (VR) [1], human-robot interaction (HRI) [2], advanced driver assistant systems (ADAS) [3], and disease prevention support systems (DPSS) [4].

Typical FER methods include three stages: (1) facial-component detection, (2) feature point extraction, and (3) facial expression classification. Facial-component detection involves extracting a facial region from an input image to obtain features such as the eyes and nose from the detected facial components. More recently, studies have shown that feature extraction can be classified into spatial [5,6], and temporal feature extraction [7]. Generally, the expression classifier and feature extraction are vital for the accuracy of FER. Many developments have been made to exploit facial expression (FE) classification, including the Bayesian Classifier [8], Hidden Markov Model (HMM) [9], Adaboost [10], and Support Vector Machine (SVM) [11]. Figure 1 shows the details of the conventional FER process.

Recent developments in deep learning have achieved significant advancements in computer vision and image processing [12,13,14,15]. Among the deep learning methods, the Convolutional Neural Network (CNN) has been proven capable of reducing the dependency on analytical models and preprocessing techniques by enabling “end-to-end” direct learning from input images. For example, feature extraction and recognition are jointly learned using deep learning methods [16,17,18].

FER is highly sensitive to intra-class variation according to age and gender, illuminance, and facial pose [19]. In addition, because FER datasets are limited and small, operating a CNN to extract the salient features that represent the facial expressions from the facial image is problematic. Several methods have been explored to overcome this problem [20]. Examples of this are the transfer learning method [21] for solving the overfitting problem in training datasets, and the ensemble architectures [22] and hybrid variant input approaches [23] for extracting discriminative features. Notably, most of these approaches primarily concentrated on designing new deep learning architectures and overlooked the loss function. Additionally, the limited training datasets remain a challenge in improving FER performance.

One method of extracting salient features from limited datasets is to change the traditional loss function of the CNN architecture to reduce the intra-class variation and increase the inter-class variation of the deep features, thereby creating discriminative features. Typically, CNN-based FER optimizes the softmax loss function, which seeks to penalize misclassified samples, encouraging the distinction of features between different classes. The softmax layer is crucial for ensuring that the learned features of various classes remain distinguishable. However, severe intra-class variation remains challenging. Advanced loss functions can be used to address this problem. Generally, advanced loss functions are divided into two categories: Angular-distance-based method (L-Softmax [24], AM-Softmax [25]), Euclidean-distance-based method (contrastive loss [26], triplet loss [27], center loss [28]).

The angular-distance-based losses have made the learned features potentially separable with a larger angular/cosine distance. These losses were reformulated based on the original softmax loss, allowing inter-class separability and intra-class compactness between learned features. However, these loss functions were difficult to converge when trained with complex datasets such as that of FER.

Furthermore, the Euclidean-distance-based losses have embedded the input images in the Euclidean space to decrease intra-class variation and increase inter-class variation. Contrastive and triplet losses increased memory load and training time owing to the complex recombination of training samples. Center loss updated the class center by reducing the distance between the deep features and their corresponding class centers. Nevertheless, it disregarded inter-class variation, thus limiting the FER performance improvement.

To summarize, the existing loss functions for CNN-based FER have the following challenges: (1) the difficulty in convergence with the complex training dataset, (2) the high memory consumption and training time, and (3) the disregard of inter-class similarity.

Given the above analysis, this study presents a variant loss to minimize the distance between the deep features and their corresponding class centers as well as maximize the distances of deep features with their non-corresponding class centers and the distances between different class centers. Figure 2 illustrates the concept of the proposed loss function. Finally, the proposed loss function was assessed on four well-known benchmark facial expression databases: the Cohn-Kanade Plus (CK+) [29], the Oulu-CASIA [30], MMI [31], and FER2013 [32] databases. The contributions of this study can be summarized as follows:

A new loss function is proposed to simultaneously consider inter- and intra-class variations, which enables CNN-based FER methods to achieve impressive performance.A new loss function can be easily optimized with various CNN architectures on diverse databases to learn the discriminative power of deep features for the FER problem.Comprehensive experiments on benchmark databases are conducted to prove that the auxiliary CNN architectures trained with the proposed loss function performed much better than with existing loss functions.

The remainder of this paper is organized into four sections. Section 2 summarizes previous loss functions and auxiliary CNN architectures. Section 3 describes the proposed loss function that simultaneously considers intra- and inter-class variations. Section 4 analyzes the simulation results, and Section 5 states the conclusions.

## 2. Related Work

### 2.1. Previous Loss Functions

The softmax loss is good at increasing the inter-class variation but cannot decrease the intra-class variation. To tackle this problem, several loss functions have been introduced to reduce the intra-class variation. Most representatively, L-Softmax loss [24] is an improvement over the conventional softmax loss, enabling inter-class separability and intra-class compactness between learned features. With an adjustable margin value, L-softmax could determine an adaptable learning task with flexible difficulty for CNNs. It also prevented overfitting problems to leverage the powerful learning capacity of deep and wide architectures. Nevertheless, when the training dataset has various subjects, the convergence of L-Softmax will be tougher than the softmax loss. AM-Softmax loss [25] used an additive margin strategy to the target logit of softmax loss with features and weights normalized. Although it was intuitively appealing and more interpretable than the L-Softmax [24], selecting the margin hyperparameter was challenging.

Contrastive [26] and triplet losses [27] adopted a pair training technique. In particular, the contrastive loss included negative and positive pairs. Its gradients attracted positive pairs and repelled negative ones. Meanwhile, triplet loss reduced the distance between an anchor and a positive sample and increased the distance between an anchor and a negative sample of a different identity. The training procedure for these losses was still challenging owing to the selection of effective training samples. Center loss [28] decreased intra-class variations during training by penalizing the distances between deep features and their corresponding class centers. By relying solely on training CNNs with center loss, the deep features and class centers might deteriorate to zero. Moreover, the center loss was minimal, and discriminative features could not be achieved. Thus, the center loss should be jointly supervised with the softmax loss during training. However, each identity’s center doubled the memory storage of the last CNN layer.

Range loss [33] was proposed to effectively use the whole long-tailed dataset in the training procedure. The range loss was optimized jointly with softmax loss as supervisory signals to train CNNs. However, the optimization strategy could be challenging because softmax loss requires uniform distribution among all the classes, and the ability to improve inter-class differences within each mini-batch was restricted. Marginal loss [34] could decrease the intra-class variances and enlarge the inter-class distances by focusing on the marginal samples. The marginal, combined with a softmax loss to supervise the learning of CNN jointly, could greatly improve the discriminative capacity of deep features for efficient facial recognition. Even so, the age variance restriction in the training data could significantly reduce the performance when there was a large year gap.

According to Table 1, while existing loss functions achieved promising performance, there is still much room for improvement. To this end, this study proposes variant loss to minimize the distance between the deep features and their corresponding class centers as well as maximize the distances of deep features with their non-corresponding class centers and the distances between different class centers. The proposed loss function is easy to adopt in CNN-based FER methods and achieves outstanding performance.

### 2.2. Auxiliary CNN Architectures

Given an input image or feature, classification models predict specific labels. In this study, six popular CNN architectures are trained using various loss functions to evaluate the feasibility of the proposed loss function. First, AlexNet [35] has eight layers comprising five convolutional layers and three fully connected layers combined with dropout techniques. Its simplicity and moderate depth made its training fast.

To improve the classification performance, InceptionNet [36] was designed based on the Inception module, which aggregated four parallel branches: three convolution branches with different kernel sizes (1 × 1, 3 × 3, and 5 × 5) and a max-pooling branch. InceptionNet contained 22 layers, including nine Inception modules stacked on top of each other. This design increased the width of the network and adaptability to various scales.

The deep learning networks also suffer from a vanishing gradient problem that impedes accuracy. ResNet [37] was proposed to add skip connections from the input to the output of the convolutional layer to address these problems. The residual block contained two 3 × 3 convolutional layers, each followed by the Batch Norm and ReLU activation function. ResNet-18 was selected to train with the comparative loss function in this study.

DenseNet [38] proposed dense blocks and transition layers. Dense blocks concatenated the output of the previous layer as the input of the next. In this way, a feed-forward nature could be maintained. However, the number of channels would be increased when concatenating layers. The transition was used to control the size of the features by 1 × 1 convolution. Moreover, the height and width of features were reduced through the average pooling layer.

Recently, MobileNetV3 [39] has been applied to mobile and embedded devices owing to its lightweight. It was based on a combination of hardware-aware network architecture search (NAS) algorithm and squeeze-and-excitation (SE) module [40]. The block-wise search algorithm (MnasNet [41]) was employed to identify global network structures, and then the layer-wise search algorithm (NetAdapt [42]) was sequentially employed to adjust individual layers. MobileNetV3 inserted the SE module to build channel-wise attention. The hard-sigmoid function was utilized to substitute the conventional sigmoid in the SE module for more efficient calculation. In addition, the hard-swish function was adopted instead of ReLU for non-linearity improvement.

Finally, ResNeSt [43] is an improved version of ResNet. It combined channel-wise attention with multi-path representation into a unified Split-Attention block. These Split-Attention blocks were stacked to follow the concept of residual learning from the ResNet model [37]. This architecture enhanced learned feature representations for multiple high-level vision tasks, including object detection, image classification, and semantic segmentation. Moreover, it was reported that ResNeSt enabled the acceleration of training and was computationally efficient.

## 3. Proposed Method

As mentioned previously, a variant loss is proposed to minimize the distance between the deep features and their corresponding class centers as well as maximize the distances of deep features with their non-corresponding class centers, and the distances between different class centers. The new loss function is expressed as follows:(1)$$L_{v}=\frac{1}{2}\sum_{i=1}^{M}(∥F\left(x_{i}\right)-c_{y_{i}}{∥}_{2}^{2}+\frac{\lambda_{1}}{\epsilon_{1}+\sum_{j=1,j\neq y_{i}}^{N}{∥F\left(x_{i}\right)-c_{j}∥}_{2}^{2}})+\frac{\lambda_{2}}{\epsilon_{2}+\sum_{m\in N}\sum_{\begin{array}{c} n\in N\\ n\neq m \end{array}}{∥c_{m}-c_{n}∥}_{2}^{2}},$$
where $y_{i}$, $x_{i}\in R^{d}$ are the ordinary label and input images of *i*-th sample facial expressions, respectively; *d* is dimension features. F(·) expresses the feature extraction from the CNNs; $c_{y_{i}}$$\in R^{d}$ denotes the $y_{i}$-th class center of the deep features from the CNNs with the same label class $y_{i}$. *M* is the number of training data in the batch size; *N* is the number of classes; $c_{j},c_{m}$, and $c_{n}\in R^{d}$ are the *j*-th, *m*-th, and *n*-th class centers of deep features, respectively. $\epsilon_{1}$ and $\epsilon_{2}$ are tolerance parameters that guarantee that the denominator is higher than zero; and $\lambda_{1}$, $\lambda_{2}$ are the hyperparameters used for balancing these loss terms.

The first term is similar to the center loss and tends to reduce the distance between the deep features and their corresponding class centers. The second and third terms tend to increase the distance between the deep features and their non-corresponding class centers and between class centers, respectively. By minimizing the proposed loss function, the intra-class variations of the deep features are reduced, whereas the inter-class variations continue to increase.

The softmax loss is obviously good at increasing the inter-class variation, and it is tractable and makes it easy to obtain the optimized solution. Therefore, the proposed loss function is applied to the batch data in each iteration to train it with the softmax loss. In addition, the most powerful networks tend to combine specific loss functions such that the supervision signals are more successfully backpropagated, mitigating the training difficulty and improving the robustness of network training [44]. In our study, the overall loss function for training the CNN is computed as the sum of the weights of the softmax and variant losses. In short, the overall loss function is expressed as follows:(2)$$L=L_{s}+\lambda L_{v},$$
where λ is a hyperparameter used for balancing the softmax and the variant losses. The overall system of the CNNs using the proposed loss is illustrated in Figure 3.

In this method, the network parameters include CNN parameters *W* and the softmax loss parameters θ are updated in mini-batches. Only the gradient of $L_{s}$ is needed to update the softmax loss parameters θ because the $L_{v}$ does not affect it. The gradient of variant loss is used to update *W*. The gradient of $L_{v}$ with respect to the $F(x_{i})$ is calculated as follows:(3)$$\begin{array}{c} \frac{dL_{v}}{dF(x_{i})}=\left(F\left(x_{i}\right)-c_{y_{i}}\right)-\lambda_{1}\frac{\sum_{j=1,j\neq y_{i}}^{N}\left(F\left(x_{i}\right)-c_{j}\right)}{{\left(\epsilon_{1}+\sum_{j=1,j\neq y_{i}}^{N}{∥F\left(x_{i}\right)-c_{j}∥}_{2}^{2}\right)}^{2}}\\ \;-\lambda_{2}\frac{\sum_{\begin{array}{c} m=y_{i}\\ n\in N\\ n\neq m \end{array}}\left(c_{y_{i}}-c_{n}\right)-\sum_{\begin{array}{c} m\in N\\ n=y_{i}\\ n\neq m \end{array}}\left(c_{m}-c_{y_{i}}\right)}{(\epsilon_{2}+\sum_{m\in N}\sum_{\begin{array}{c} n\in N\\ n\neq m \end{array}}∥c_{m}-c_{n}{{∥}_{2}^{2})}^{2}}. \end{array}$$

In addition, the class center is calculated by averaging the features in the same class and updating in each iteration. The centers are updated as follows:(4)$$c_{k}^{t+1}=c_{k}^{t}+\alpha \nabla c_{k},$$
where α is the learning rate of class centers.

The update of the *k*-th class center can be computed as a derivative of the variant loss with respect to the class center $c_{k}$:(5)$$\begin{array}{c} \nabla c_{k}=\frac{\sum_{\begin{array}{c} i=1 \end{array}}^{M}\delta \left(y_{i},k\right)\left(c_{k}-F\left(x_{i}\right)\right)}{1+\sum_{\begin{array}{c} i=1 \end{array}}^{M}\delta \left(y_{i},k\right)}-\lambda_{1}\frac{\sum_{\begin{array}{c} i=1 \end{array}}^{M}\left(1-\delta \left(y_{i},k\right)\right)\left(c_{k}-F\left(x_{i}\right)\right)}{(\epsilon_{1}+\sum_{j=1,j\neq y_{i}}^{N}||F\left(x_{i}\right)-c_{j}{\left|\right|}_{2}^{2}{)}^{2}}\\ \;-\lambda_{2}\frac{\sum_{m\in N}\delta \left(m,k\right)\sum_{\begin{array}{c} n\in N\\ n\neq m \end{array}}\left(c_{k}-c_{n}\right)}{(\epsilon_{2}+\sum_{m\in N}\sum_{\begin{array}{c} n\in N\\ n\neq m \end{array}}∥c_{m}-c_{n}{{∥}_{2}^{2})}^{2}\left(1+\sum_{m\in N}\delta \left(m,k\right)\right)}, \end{array}$$
where $\delta (y_{i},k)$ and δ(m,k) are defined as
(6)$$\delta \left(y_{i},k\right)=\left\{\begin{array}{c} 1;y_{i}=k\\ 0;y_{i}\neq k \end{array}\right.,$$
(7)$$\delta \left(m,k\right)=\left\{\begin{array}{c} 1;m=k\\ 0;m\neq k \end{array}\right..$$

CNNs can be trained utilizing standard stochastic gradient descent (SGD) [45]. The hyperparameters of CNNs contain a batch size *M*, the number of training iterations *T*, the learning rates of the weight parameter μ, the learning rates of the class centers α, and balanced terms of the loss function λ, $\lambda_{1}$, $\lambda_{2}$. First, the parameters of the CNNs are initialized *W*, the softmax loss parameters θ, and class center $c_{k}$. In each iteration, *M* training images $x_{i}$ are passed into the CNNs to obtain the output of the last fully-connected layer $F(x_{i})$ in each batch. The overall loss of the model and derivative of the loss functions with respect to the output of the last fully-connected layers $F(x_{i})$ are calculated to update the parameters of the CNNs. θ is independent of the variant loss; therefore, only the softmax loss is considered. Furthermore, the gradient of variant loss is used to update *W*. The update process of *W* and θ are separated with different derivatives. Finally, the derivative of the variant loss with respect to class center $c_{k}$ is calculated to update the class center $c_{k}$ with the learning rate of the class centers α. The training process for CNNs with proposed loss is interpreted in Algorithm 1.
**Algorithm 1** Training process for CNNs with proposed loss     **Input:** Training images $x_{i}$, batch size *M*, number of training iterations *T*, learning rates of weight parameter μ, learning rate of class centers α, hyper-parameters λ, $\lambda_{1}$, $\lambda_{2}$.     **Initialization:** the CNNs parameters *W*, the softmax loss parameters θ, the class centers $c_{k}$, the iteration *t* = 0.1: **while**
 t≤T 
**do**2:    Calculate the deep features, the output of the last fully-connected layers $F(x_{i})$ of *M* input images in one mini-batch.3:    Calculate the overall loss as in (Equation 2):4:    $L=L_{s}+\lambda L_{v}$5:    Calculate the gradients for each input *i* by:6:    $\frac{dL^{t}}{dF{\left(x_{i}\right)}^{t}}=\frac{dL_{s}^{t}}{dF{\left(x_{i}\right)}^{t}}+\lambda \frac{dL_{v}^{t}}{dF{\left(x_{i}\right)}^{t}}$7:    Update parameters θ by:8:    $\theta^{t+1}$ = $\theta^{t}-\mu \frac{dL^{t}}{d\theta^{t}}=\theta^{t}-\mu \frac{dL_{s}^{t}}{d\theta^{t}}$9:    Update parameters *W* by:10:    $W^{t+1}=W^{t}-\mu \frac{dL^{t}}{dW^{t}}=W^{t}-\mu \sum_{\begin{array}{c} i \end{array}}^{M}\frac{dL^{t}}{dF{\left(x_{i}\right)}^{t}}\frac{dF{\left(x_{i}\right)}^{t}}{dW^{t}}$11:    Update $c_{k}$ for *k*-th class center: $c_{k}^{t+1}=c_{k}^{t}-\alpha \nabla c_{k}$12:    t=t+113:**end while**     **End of the algorithm:** The CNNs parameters *W*, the softmax loss parameters θ

## 4. Experiments

### 4.1. Experimental Setup

The performance of the proposed method was evaluated based on four benchmark facial expression databases: three from a laboratory environment, namely, Cohn-Kanade Plus (CK+) [29], Oulu-CASIA [30], and MMI [31]; and one from a wild environment, FER2013 [32]. A 10-fold cross-validation strategy was employed for model evaluation, especially focusing on scenarios with small and imbalanced datasets, such as CK+, MMI, and Oulu-CASIA. The amount of data for training depends on several factors, such as the task’s complexity, the data’s diversity, the desired output, the data quality, and the deep model architecture. In this study, each of these databases was strictly divided into 90% as a training set and 10% allocated as a testing set. Furthermore, FER is a large-scale dataset; the training and evaluation processes were conducted on its provided datasets. To prevent overfitting issues, we carefully chose the appropriate weight based on the learning process of the model to achieve a satisfactory performance. Several sample images derived from these databases are illustrated in Figure 4. The details of the databases and the number of images for each emotion are presented in Table 2.

To minimize the variations in the face scale and in-plane rotation, the face was detected and aligned from the original database using the OpenCV library with Haar–Cascade detection [46]. The aligned facial images were resized to 64 × 64 pixels. Moreover, intensity equalization was used to enhance the contrast in facial images. A data augmentation technique was used to overcome the restricted number of training images in the FER problem. Furthermore, the facial images were flipped, and each one and its corresponding flipped image was rotated at −15, −10, −5, 5, 10, and 15°. The training databases were augmented 14 times using original, flipped, six-angle, and six-angle-flipped images. The rotated facial images are shown in Figure 5.

The proposed loss function was compared with softmax, center [28], range [33], and marginal losses [34] using the same CNN architectures to demonstrate the effectiveness of the proposed loss function. Accuracy is a crucial quantitative metric to evaluate the performance of the proposed method, which can be calculated as follows:(8)$$Accuracy=\frac{\mathrm{Number}\mathrm{of}\mathrm{correct}\mathrm{predictions}}{\mathrm{Total}\mathrm{number}\mathrm{of}\mathrm{predictions}}.$$

The experiment was conducted in a subject-independent scenario. The CNN architectures were processed with 64 images in each batch. The training was performed using the standard SGD technique to optimize the loss functions. The hyper-parameter λ was used to balance the softmax and variant losses. $\lambda_{1}$ and $\lambda_{2}$ were utilized to balance among these losses in the variant loss, and α controlled the learning rate of the class center $c_{k}$. All of these factors affect the performance of our model. In this experiment, the values λ = 0.001, $\lambda_{1}$ = 0.4, $\lambda_{2}$ = 0.6, $\epsilon_{1}$ = $\epsilon_{2}$ = 0.001 were empirically selected for the proposed loss. For the center, marginal, and range losses, λ was set to 0.001. The detailed specifications of the implemented environment are shown in Table 3.

### 4.2. Experimental Results

(1) Results on Cohn-Kanade Plus (CK+) database: The CK+ is a representative laboratory-controlled database for FER. It comprises 593 image sequences collected from 123 participants. A total of 327 of these image sequences have one of seven emotion labels: anger, contempt, disgust, fear, happiness, sadness, and surprise, from 118 subjects. Each image sequence starts with a neutral face and ends with the peak emotion. To collect additional data, the last three frames of each sequence were collected and associated with the provided labels. Therefore, a database containing 981 experimental images was constructed. The images were primarily grayscale and digitized to a 640 × 490 or 640 × 480 resolution.

The average recognition precision of the methods based on the loss functions and CNN architectures is listed in Table 4. The accuracy of the proposed loss function was superior to that of the others for all six CNN architectures. For the same loss functions, the accuracy of ResNet was the highest, followed by those of MobileNetV3, ResNeSt, InceptionNet, AlexNet, and DenseNet. Overall, the proposed loss produced an average recognition accuracy of 94.89% for the seven expressions using ResNet.

Table 5 presents the confusion matrix [47] of the ResNet, which was optimized using the proposed loss function. The accuracy of the contempt, disgust, happiness, and surprise labels was significant. Notably, the happiness percentage was the highest at 99.5%, followed closely by surprise, disgust, and contempt at 98.4%, 97.7%, and 93.4%, respectively. The proportions of the anger, fear, and sadness labels were inferior to these emotions because of their visual similarity.

A receiver operating characteristic (ROC) curve [48] and the corresponding area under the curve (AUC) for all expression recognition performances are illustrated in Figure 6. An increase in the AUC signifies an improved ability of the model to differentiate between various classes. The value of the disgust, happiness, and surprise labels reach peak values at 100%. The others also gained a relatively high classified range of 97%, 94%, 89%, and 86% for corresponding emotional classes anger, sadness, fear, and contempt.

(2) Results on Oulu-CASIA database: The Oulu-CASIA database includes 2880 image sequences obtained from 80 participants using a visible light (VIS) imaging system under normal illumination conditions. Six emotion labels were assigned to each image sequence: anger, disgust, fear, happiness, sadness, and surprise. Like the Cohn-Kanade Plus database, the image sequence started with a neutral face and ended with the peak emotion. For each image sequence, the last three frames were collected as the peak frames of the labeled expression. The imaging hardware was operated at 25 fps with an image resolution of 320 × 240 pixels.

The average recognition accuracy of the methods is listed in Table 6. The performance of the proposed loss function was comparable to that of previous ones. Specifically, the proposed loss function achieved an average recognition accuracy of 77.61% for the six expressions using the ResNet architectures.

Table 7 presents the confusion matrix of ResNet trained with the proposed loss function. The accuracy of the happiness and surprise labels increased, with the former achieving 92.1% and the latter gaining 84.0%. The accuracy for anger, disgust, fear, and sadness was inferior, obtaining 66.2%, 70.5%, 76.3%, and 76.5%, respectively.

Figure 7 shows a receiver operating characteristic (ROC) curve, which verifies the performance of a recognition model. The range for all emotional labels was relatively significant. Among them, the result of the happiness, surprise, and sadness class illustrates the AUC over 90%, followed by disgust, fear, and anger at 89%, 86%, and 85%, respectively.

(3) Results on MMI database: The laboratory-controlled MMI database comprises 312 image sequences collected from 30 participants. A total of 213 image sequences were labeled with six facial expressions: anger, disgust, fear, happiness, sadness, and surprise. Moreover, 208 sequences from 30 participants were captured in frontal view. The spatial resolution was 720 × 576 pixels, and the videos were recorded at 24 fps. Unlike the Cohn-Kanade Plus and Oulu-CASIA databases, the MMI database features image sequences labeled by the onset-apex. Therefore, the sequences started with a neutral expression, peaked near the middle, and returned to a neutral expression. The location of the peak expression frame was not provided. Furthermore, the MMI database presented challenging conditions, particularly in the case of large interpersonal variations. Three middle frames were chosen as the peak expression frames in each image sequence to conduct a subject-independent cross-validation scenario.

Table 8 lists the average recognition accuracy of the methods. Our loss function outperformed all the other loss functions by a certain margin. Specifically, the proposed loss function achieved average recognition accuracy of 67.43% for the six expressions using the MobileNetV3 architecture.

Table 9 presents the percentages in the confusion matrix of the MobileNetV3 optimized with the proposed loss function. The accuracy for all emotions was under 80.0%, except for happiness and surprise, which obtained 89.7% and 81.3%, respectively. This may be due to the number of images in each class. An instance of this is fear, which had the fewest labels and whose accuracy was a low 31.0%. Similar results were also confirmed for the accuracy of anger, disgust, and sadness.

A ROC curve and corresponding AUC for all facial expression recognition performances are presented in Figure 8. The value of the fear and anger classes is lowest, with the former achieving 69% and the latter 79%. The value for sadness, disgust, surprise, and happiness was higher, acquiring 83%, 88%, 89%, and 98%, respectively.

(4) Results on FER2013 database: FER2013 is a large-scale, unconstrained database automatically collected by the Google image search API. It includes 35,887 images with a relatively low resolution of 48 × 48 pixels, which are labeled with one of seven emotion labels: anger, disgust, fear, happiness, sadness, surprise, and neutral. The training set comprises 28,709 examples. The public test set consists of 3589 examples; the remaining 3589 images are used as a private test set.

Table 10 lists all the methods’ average recognition accuracy. The accuracy of the proposed loss function greatly exceeds that of the others in all CNN architectures, except AlexNet. The proposed loss function achieved a peak average recognition accuracy of 61.05% for the seven expressions using the ResNeSt architecture.

The confusion matrix of ResNeSt, which was trained with the proposed loss function, is presented in Table 11. The happiness percentage was highest at 80.7%, followed by surprise at 77.4%. The others obtained relatively low prediction ratios.

Figure 9 depicts a ROC curve, where the range for all emotions was over 70%, except for sadness at 69%. Among other expression classes, the result of the surprise class illustrates the highest AUC value at 92%, followed by happiness, disgust, anger, fear, and neutral at 88%, 82%, 74%, 72%, and 72%, respectively.

### 4.3. Training Time

The training time is essential for evaluating the computational complexity of deep learning networks with specific loss functions. This section compares the training time of the auxiliary CNN architectures with the existing and proposed loss functions. Notably, all loss functions were trained on a single GPU. Depending on the dataset and network architecture, the number of iterations was empirically set to achieve optimal convergence with the corresponding loss function. When the data have been pre-processed, we start measuring the training time T = T1 − T0 with the beginning time T0 at the start of the first iteration and ending time T1 at the finish of the final iteration. As presented in Table 12, the softmax loss trained the fastest because it only uses one term in the mathematical function, followed closely by the center and proposed loss functions. Furthermore, the range and marginal loss functions required longer training times among the compared methods because their complex mathematical functions produced a time-consuming backpropagation process. In summary, only softmax and center loss were marginally faster than the proposed method. However, the proposed method achieved superior performance compared with these loss functions. Therefore, the proposed method is computationally efficient and meets the practical requirements.

To summarize, the computational cost of the loss function in deep learning is critical. The loss function is used for evaluation during training, so a computationally expensive loss function slows down the training process and can cause bottlenecks, especially for large datasets. In addition, designing the loss function depends on the purpose of the output. Therefore, a good trade-off between computational cost and accuracy is desired.

## 5. Conclusions

Although a loss function can drive network learning, it has received little attention for promoting facial expression recognition (FER) performance. This study presents a new loss function that allows simultaneous consideration of inter- and intra-class variations to be applied to CNN architecture for FER. More specifically, this loss function minimizes the distance between the deep features and their corresponding class centers as well as maximizes the distances of deep features with their non-corresponding class centers and the distances between different class centers. In addition, the proposed loss function improves the testing accuracy of the benchmark FER database compared with several other loss functions. Overall, this study demonstrates that choosing optimal loss functions strongly affects the performance of deep learning networks, even when maintaining their architecture. While the proposed loss function achieved impressive performance, it has not completely solved the unbalanced data problems. To overcome this issue, we plan to apply resampling methods by undersampling with majority class samples, oversampling with minority class samples, and using cost-sensitive learning to focus on the minority classes. In addition, we would like to extend the experiment for facial expression recognition in real-time conditions with a variety of emotions (e.g., embarrassment, adoration, nostalgia, satisfaction, pride, etc.). Currently, the proposed loss function applied to facial expression recognition for masked faces is under investigation and is expected to achieve promising performance.

## Figures and Tables

**Figure 1 sensors-23-09658-f001:**
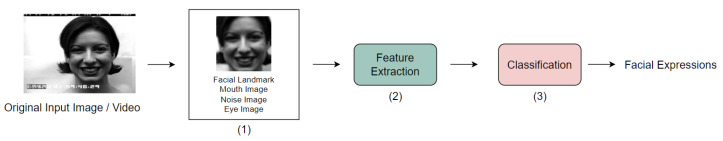
Pipeline of the FER system.

**Figure 2 sensors-23-09658-f002:**
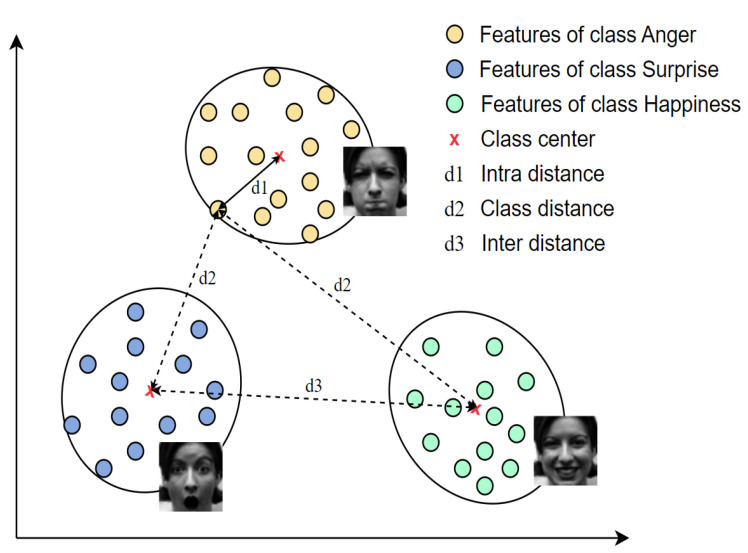
Visualization of the proposed loss for one batch image in the Euclidean space. Supposing the three classes anger, happiness, and surprise in this batch, the proposed loss function aims to reduce the intra-distance d1 and enhance the inter-distance d2 and class distance d3 (Best viewed in color).

**Figure 3 sensors-23-09658-f003:**
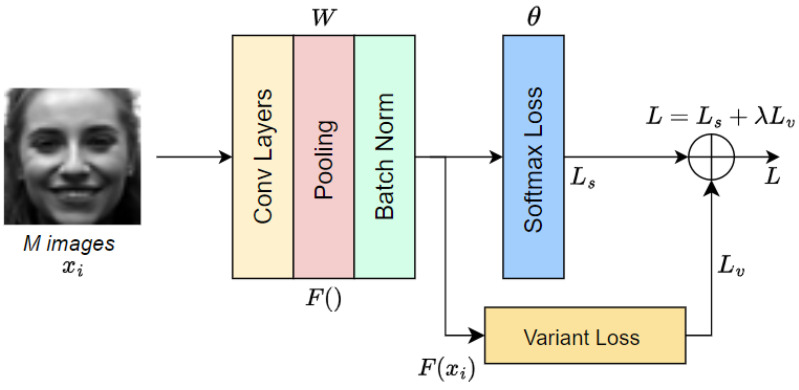
Overall system CNNs using the proposed loss function.

**Figure 4 sensors-23-09658-f004:**
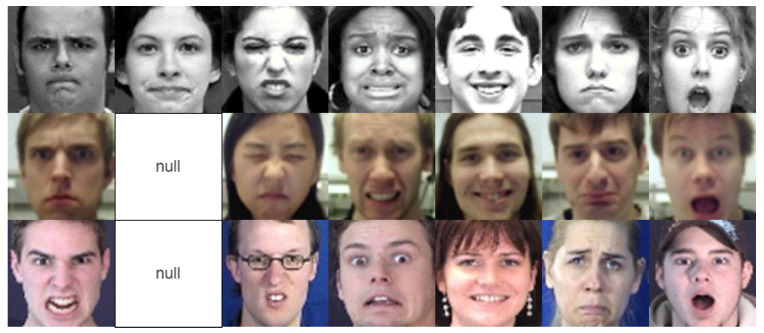
Example face images from CK+ (**top**), Oulu-CASIA (**center**), and MMI (**bottom**) databases. The facial expressions from left to right convey anger, contempt, disgust, fear, happiness, sadness, and surprise. The contempt images of Oulu-CASIA and MMI are null.

**Figure 5 sensors-23-09658-f005:**
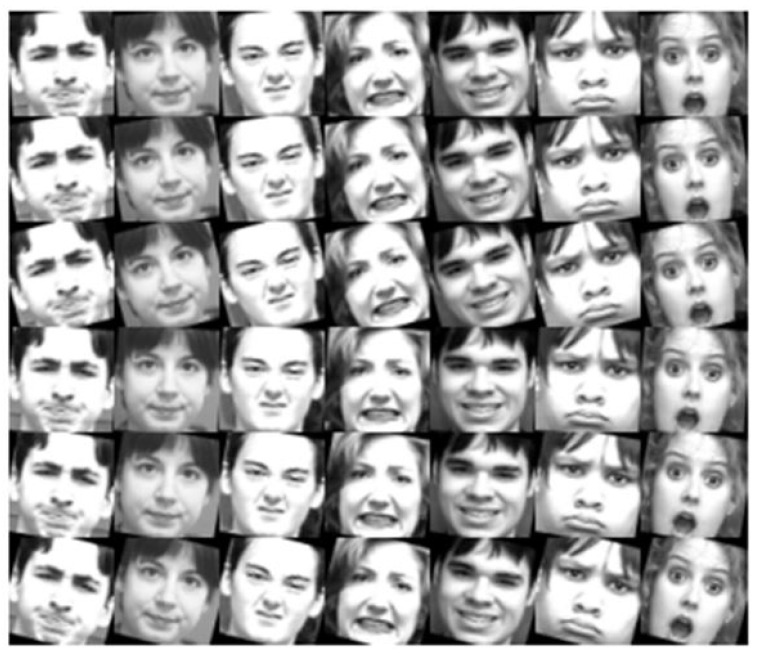
Example rotated images from CK+ database. The facial expressions from left to right convey anger, contempt, disgust, fear, happiness, sadness, and surprise. The rotation degrees from top to bottom are −5, −10, −15, 5, 10, 15°.

**Figure 6 sensors-23-09658-f006:**
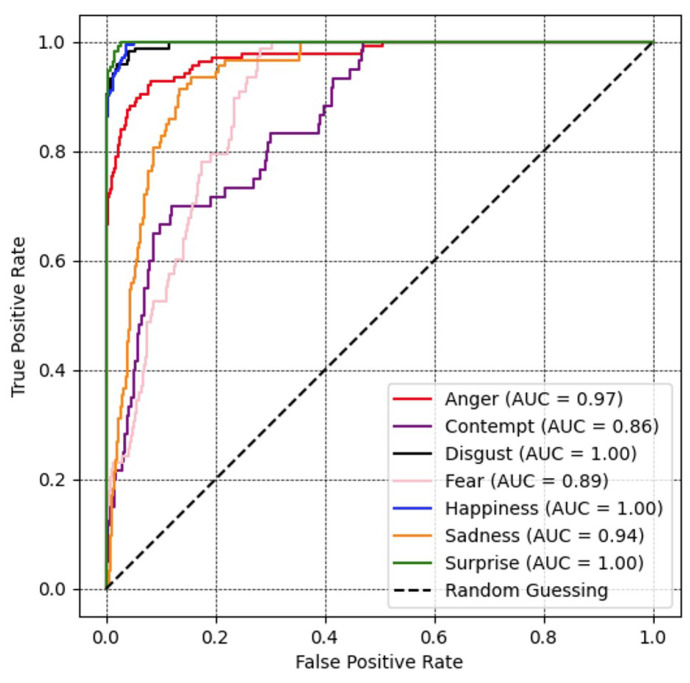
Recognition performance portrayed as ROC curves and corresponding area under the curve (AUC) for all expression recognition performances with ResNet optimized with the proposed loss on the CK+ database.

**Figure 7 sensors-23-09658-f007:**
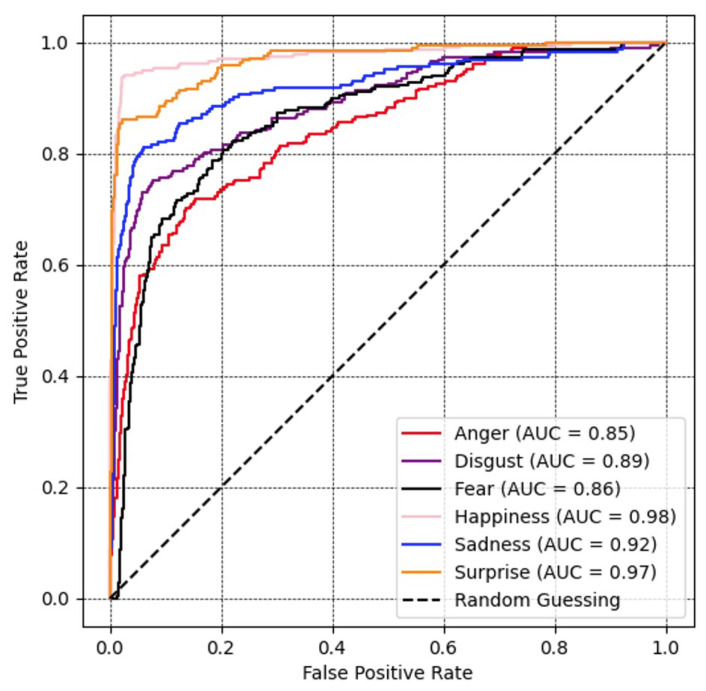
Recognition performance portrayed as ROC curves and corresponding area under the curve (AUC) for all expression recognition performances with ResNet optimized with the proposed loss function on the Oulu-CASIA database.

**Figure 8 sensors-23-09658-f008:**
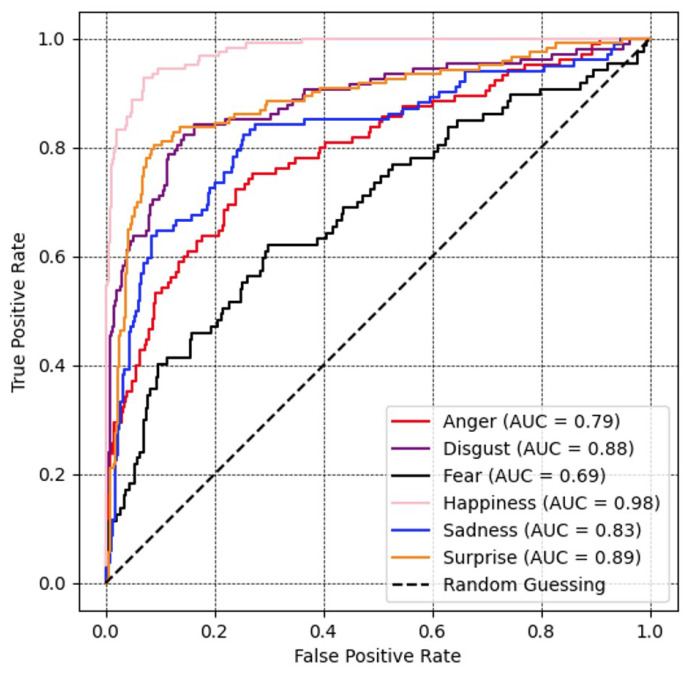
Recognition performance portrayed as ROC curves and corresponding area under the curve (AUC) for all expression recognition performances with MobileNetV3 optimized with the proposed loss on the MMI database.

**Figure 9 sensors-23-09658-f009:**
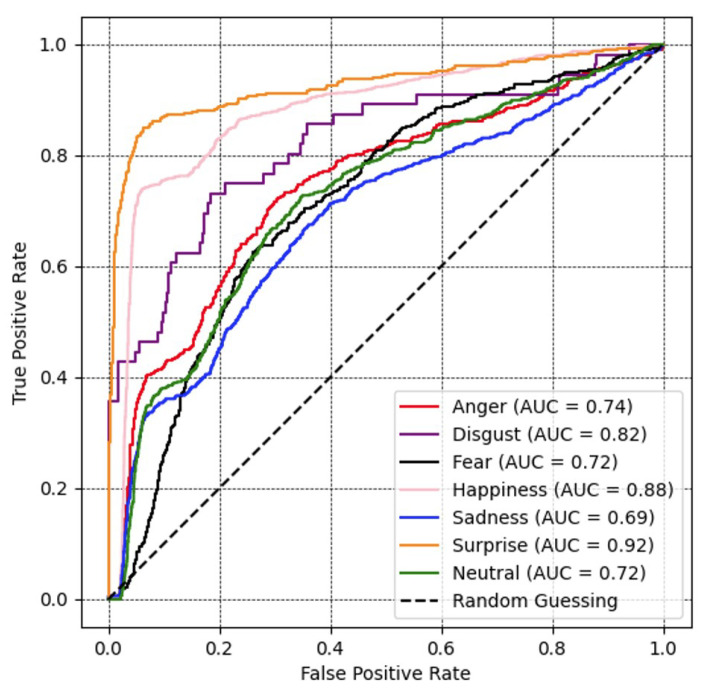
Recognition performance portrayed as ROC curves and corresponding area under the curve (AUC) for all expression recognition performances with ResNeSt optimized with the proposed loss function on the FER2013 database.

**Table 1 sensors-23-09658-t001:** The properties of previous loss functions in deep facial recognition.

Loss Functions	Consider Intra-Class Variation	Consider Inter-Class Variation	Limitations
L-Softmax [24]	No	Yes	The convergence is challenging
AM-Softmax [25]	No	Yes	The hyperparameter selection is challenging
Contrastive [26]	Yes	Yes	The convergence is challenging
Triplet [27]	Yes	Yes	The convergence is challenging
Center [28]	Yes	No	A large memory storage is required
Range [33]	Yes	Yes	A optimization strategy is challenging
Marginal [34]	Yes	Yes	A optimization strategy is challenging

**Table 2 sensors-23-09658-t002:** Number of images for each emotion: anger (An), contempt (Co), disgust (Di), fear (Fe), happiness (Ha), sadness (Sa), surprise (Su), neutral (Ne).

	An	Co	Di	Fe	Ha	Sa	Su	Ne	All
CK+	135	54	177	75	207	84	249	-	981
Oulu	240	-	240	240	240	240	240	-	1440
MMI	99	-	96	84	126	96	123	-	624
FER2013	4953	-	547	5121	8989	6077	4002	6198	35,887

**Table 3 sensors-23-09658-t003:** Configuration information of the experimental environment.

Experimental Environment	Configuration Parameters
CPU	Intel${}^{®}$Xeon${}^{®}$ CPU E5-2620 v2, 48 GB RAM
GPU	NVIDIA GeForce RTX 3090
Operating system	Ubuntu 22.04
Deep learning framework	Pytorch 1.13.1
Programming language	Python 3.10

**Table 4 sensors-23-09658-t004:** Performance comparison on the CK+ database in terms of the seven expressions.

Method	AlexNet	InceptionNet	ResNet	DenseNet	MobileNetV3	ResNeSt
Softmax	87.38	87.18	90.65	83.68	91.60	85.58
Center	88.08	87.88	92.46	83.38	87.78	85.98
Range	90.59	88.08	91.79	85.28	91.50	88.68
Marginal	89.18	86.68	87.78	84.18	89.68	86.38
Proposed	**90.79**	**89.18**	**94.89**	**85.98**	**91.90**	**89.28**

**Table 5 sensors-23-09658-t005:** Confusion matrix of ResNet optimized with the proposed loss on the CK+ database. The labels in the leftmost column and on top represent the ground truth and the prediction results, respectively.

	An	Co	Di	Fe	Ha	Sa	Su
An	**86.2%**	1.4%	6.5%	0%	0%	5.1%	0.8%
Co	3.6%	**93.4%**	0%	1.5%	0%	1.5%	0%
Di	1.7%	0%	**97.7%**	0%	0.6%	0%	0%
Fe	0%	2.6%	0%	**87.2%**	7.6%	2.6%	0%
Ha	0%	0%	0%	0.5%	**99.5%**	0%	0%
Sa	7.5%	1.1%	1.1%	0%	0%	**90.3%**	0%
Su	0.8%	0%	0%	0.4%	0%	0.4%	**98.4%**

**Table 6 sensors-23-09658-t006:** Performance comparison on the Oulu-CASIA database in terms of the six expressions.

Method	AlexNet	InceptionNet	ResNet	DenseNet	MobileNetV3	ResNeSt
Softmax	70.52	65.16	72.46	68.67	73.24	70.23
Center	71.95	64.09	74.96	69.09	74.89	67.23
Range	72.17	64.38	74.11	69.52	69.09	63.80
Marginal	70.24	68.09	71.88	68.67	73.74	68.95
Proposed	**72.96**	**69.17**	**77.61**	**69.88**	**76.46**	**70.24**

**Table 7 sensors-23-09658-t007:** Confusion matrix of ResNet optimized with the proposed loss function on the Oulu-CASIA database. The labels in the leftmost column and on top represent the ground truth and the prediction results, respectively.

	An	Di	Fe	Ha	Sa	Su
An	**66.2%**	13.0%	6.9%	0%	13.9%	0%
Di	12.4%	**70.5%**	7.3%	2.6%	6.8%	0.4%
Fe	5.8%	1.3%	**76.3%**	5.4%	5.0%	16.3%
Ha	0%	2.1%	5.8%	**92.1%**	0%	0%
Sa	12.4%	3.8%	5.1%	1.7%	**76.5%**	0.4%
Su	1.4%	0%	11.9%	2.7%	0%	**84.0%**

**Table 8 sensors-23-09658-t008:** Performance comparison on the MMI database in terms of the six expressions.

Method	AlexNet	InceptionNet	ResNet	DenseNet	MobileNetV3	ResNeSt
Softmax	57.76	53.92	61.59	60.52	61.44	54.07
Center	58.98	58.52	61.92	59.29	64.36	57.29
Range	62.67	61.75	61.13	54.68	64.20	55.76
Marginal	59.44	55.14	57.62	57.61	64.66	53.00
Proposed	**63.13**	**63.74**	**65.89**	**61.13**	**67.43**	**58.83**

**Table 9 sensors-23-09658-t009:** Confusion matrix of MobileNetV3 optimized with the proposed loss function on the MMI database. The labels in the leftmost column and on top represent the ground truth and prediction results, respectively.

	An	Di	Fe	Ha	Sa	Su
An	**57.1%**	14.3%	11.4%	3.8%	12.4%	1.0%
Di	13.0%	**72.2**%	2.8%	4.6%	4.6%	2.8%
Fe	11.5%	5.8%	**31.0%**	9.2%	11.5%	31.0%
Ha	0%	6.3%	1.6%	**89.7%**	0%	2.4%
Sa	14.7%	13.8%	8.8%	0%	**59.8%**	2.9%
Su	4.1%	0.8%	7.3%	1.6%	4.9%	**81.3%**

**Table 10 sensors-23-09658-t010:** Performance comparison on the FER2013 database in terms of the seven expressions.

Method	AlexNet	InceptionNet	ResNet	DenseNet	MobileNetV3	ResNeSt
Softmax	**59.77**	55.92	59.21	59.15	56.33	60.85
Center	58.48	57.42	56.70	59.82	50.43	60.93
Range	58.65	56.22	48.37	59.59	52.99	60.96
Marginal	59.04	57.12	57.51	58.71	56.56	59.76
Proposed	58.51	**57.81**	**59.65**	**60.46**	**58.29**	**61.05**

**Table 11 sensors-23-09658-t011:** Confusion matrix of ResNeSt optimized with the proposed loss function on the FER2013 database. The labels in the leftmost column and on top represent the ground truth and prediction results, respectively.

	An	Di	Fe	Ha	Sa	Su	Ne
An	**55.7%**	0.4%	8.6%	6.6%	14.6%	3.2%	10.9%
Di	23.2%	**46.4%**	7.2%	1.8%	10.7%	3.6%	7.1%
Fe	8.9%	0.2%	**42.5%**	4.2%	23.4%	8.3%	12.5%
Ha	3.6%	0%	1.5%	**80.7%**	3.6%	2.8%	7.8%
Sa	13.6%	0.5%	11.8%	6.9%	**48.5%**	2.8%	15.9%
Su	4.3%	0%	7.9%	3.9%	2.4%	**77.4%**	4.1%
Ne	9.9%	0.2%	7.1%	8.7%	16.8%	2.3%	**55.0%**

**Table 12 sensors-23-09658-t012:** Training time(s) comparison of the auxiliary CNN architecture with different loss functions.

**Methods**	**AlexNet**				**InceptionNet**				**DenseNet**			
	**CK+**	**Oulu-CASIA**	**MMI**	**FER2013**	**CK+**	**Oulu-CASIA**	**MMI**	**FER2013**	**CK+**	**Oulu-CASIA**	**MMI**	**FER2013**
Softmax	146	142	116	150	499	497	479	1529	393	504	394	2036
Center	150	143	121	157	502	493	494	1621	418	494	400	2050
Range	2871	2937	2219	2619	3432	3265	3219	9238	2679	3231	2460	12,113
Marginal	15,424	15,552	12,149	15,222	15,625	15,621	15,513	46,090	12,443	15,717	12,372	62,438
Proposed	157	156	127	163	520	518	522	1677	410	503	412	2177
Iterations	10,000	10,000	8000	10,000	10,000	10,000	10,000	30,000	10,000	10,000	8000	10,000
**Methods**	**ResNet**				**MobileNetV3**				**ResNeSt**			
	**CK+**	**Oulu-CASIA**	**MMI**	**FER2013**	**CK+**	**Oulu-CASIA**	**MMI**	**FER2013**	**CK+**	**Oulu-CASIA**	**MMI**	**FER2013**
Softmax	231	252	254	568	431	1144	1066	3215	2008	1619	2110	4108
Center	235	265	250	600	373	1055	1195	2415	2466	1779	2200	10,969
Range	2361	2533	2443	5774	5247	6445	6176	14,283	7368	5936	7856	64,143
Marginal	11,648	12,632	11,648	30,686	12,164	34,076	33,580	83,618	37,944	29,172	35,354	197,333
Proposed	243	271	266	601	456	1072	1108	2943	2570	1780	2273	14,004
Iterations	7500	8000	8000	20,000	7500	20,000	20,000	50,000	20,000	15,000	20,000	100,000

## Data Availability

Data are contained within the article.

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
