# Peer review of "CNN-Based Facial Expression Recognition with Simultaneous Consideration of Inter-Class and Intra-Class Variations"

_sensors, 2023, doi:10.3390/s23249658_

Round 1

Reviewer 1 Report

Comments and Suggestions for Authors

The performances achieved using the proposed function when applied to the CK database are notable. The algorithmic part is very clear and accessible.

Some minor points: from the text it clearly emerges that the tested architecture implements the combination of the proposed loss function (Variant Loss Function, L_v) with the Softmax Loss Function (c.f. eq.2); However, It is not well explained the reason why it was decided to make this sum as a general rule for this Overall Loss Function (i.e. L) and why the Softmax Loss Function was chosen among all those presented in the state of the art. The manuscript quality will be increased by deeply explaining these aspects.

Coming to the results, the architecture with integrated property loss function was tested on 4 static databases, showing notable performance for the CK database. Differently, with the MMI database, where the images are collected in challenging conditions, the performances seems to  be limited and in any case not significantly better than those achievable with the standard loss functions. For this reason it could be useful to recommend testing the proposed function also in real-time conditions on active subjects. Authors should also simply comments on this.

Finally, a last notable point is the computational superiority of the proposed function, given that typically these types of systems have such high computational costs as to discourage their real-time use.

Last considerations: I noticed a slight inconsistency between the wording in the text and the equations reported in the "Proposed Method" section, please, control and revise.

Some examples of this:

- lines 196 and 201: the use of the gradient and the derivative of the Overall Loss Function (i.e. L, as per equation 2) is introduced to calculate the parameters M and gradiente_c_k, respectively. However, in equations 3 and 4 it is shown that the gradient and the derivative are carried out by L_v or the "Variant Loss Function" rather than by L.

- Same discrepancy between lines 213-216, where a recap is carried out of the method and the reported algorithm

Comments on the Quality of English Language

no comment

Reviewer 2 Report

Comments and Suggestions for Authors

1.     The manuscript deals with deep learning based recognition of facial expressions.  The authors have experimented with multiple benchmarked databases and attempted to recognize eight types of expressions.

2.     The manuscript is very structured, well-written and comprehensive.

3.     It is highly recommended that the authors present a ‘table’ of comparison of the proposed work with similar research works and state-of-the-art research studied as part of the literature review.  This will go a long way in emphasizing the research gaps and highlighting the specific contributions of the proposed work.  Irrespective of the presented discussion, this should be done in terms of contrast and comparison of approach as well as the contrast and comparison of the results.

4.     Either human face identifiable images should not be used, or appropriate disclaimers and/or permissions of the concerned human subjects must be taken care of appropriately.

5.     Authors have not experimented with all possible types of emotions (embarrassment, adoration, nostalgia, satisfaction, pride, to name a few).  It would be interesting to know the comments of the authors on this.

Reviewer 3 Report

Comments and Suggestions for Authors

CNN-based Facial Expression Recognition with Simultaneous

Consideration of Inter-class and Intra-class Variations

This paper is well-written but requires some major improvements, which are

mentioned below:

1. Please, explain the proposed method in more detail. What is the novelty of the proposed method compared to the state of the art? Please make the flowchart of the proposed method more extended.

2. The introduction section is weak, which makes it difficult to identify novel points in Facial Expression, especially those focused on challenges, what type of challenges?

3. Any way to reduce the computational cost compared to other approaches, please discuss.

4. More information is required about the method followed in the so-called subjective evaluation. I mean about the procedure and environment (the information provided to the subjects.

5. The authors didn’t mention any limitations of the proposed method, they should. And future research direction on how to fix that challenge.

6. The main contributions of the paper should be mentioned in the Introduction section.

7. Can the authors comment on the minimal requirement for the training data set (e.g., the number of images required)? Are you using the augmentation technique to increase the training dataset artificially?

8. I haven’t seen any visual output results of the compared methods, please compare them with the proposed method.

9. Why does the paper not include any mathematical expressions such as equations or confusion matrices? Then how the authors calculate the image operations and accuracy of the method.

10. Proposed method can detect Facial Expression if the children have the masks on the face? Please check the following paper and include the results https://doi.org/10.3390/s22228704.

11. Achieved results caused by overfitting? Please discuss this in the experiment section.

12. Please make a Table in the Experimental Results section to show detailed specifications of the implemented environment such as operating system, programming language, GPU, CPU server, etc.

13. Comparing the performance based on the golden ratio Φ or drawing the ROC curves will be very useful for readers.

14. Conclusion should highlight the achievements. Here it is more or less similar to the Abstract.

Comments on the Quality of English Language

Extensive editing of English language required

Round 2

Reviewer 3 Report

Comments and Suggestions for Authors

Can be accepted after minor English corrections 

Comments on the Quality of English Language

Can be accepted after minor English corrections